# A post-translational modification of human Norovirus capsid protein attenuates glycan binding

Alvaro Mallagaray[1], Robert Creutznacher[1], Jasmin Dülfer [2], Philipp H.O. Mayer[3], Lena Lisbeth Grimm[1], Jose Maria Orduña[1,6], Esben Trabjerg[4,7], Thilo Stehle[3], Kasper D. Rand [4], Bärbel S. Blaum[3], Charlotte Uetrecht [2,5] & Thomas Peters [1]

Attachment of human noroviruses to histo blood group antigens (HBGAs) is essential for infection, but how this binding event promotes the infection of host cells is unknown. Here, we employ protein NMR experiments supported by mass spectrometry and crystallography to study HBGA binding to the P-domain of a prevalent virus strain (GII.4). We report a highly selective transformation of asparagine 373, located in an antigenic loop adjoining the HBGA binding site, into an iso-aspartate residue. This spontaneous post-translational modification (PTM) proceeds with an estimated half-life of a few days at physiological temperatures, independent of the presence of HBGAs but dramatically affecting HBGA recognition. Sequence conservation and the surface-exposed position of this PTM suggest an important role in infection and immune recognition for many norovirus strains.

---

[1] Institute of Chemistry and Metabolomics, Center of Structural and Cell Biology in Medicine (CSCM), University of Lübeck, Ratzeburger Allee 160, 23562 Lübeck, Germany. [2] Heinrich Pette Institute, Leibniz Institute for Experimental Virology, Martinistrasse 52, 20251 Hamburg, Germany. [3] Interfaculty Institute of BiochemistryUniversity of Tübingen, University of Tübingen, Hoppe-Seyler-Strasse 4, 72076 Tübingen, Germany. [4] Department of Pharmacy, University of Copenhagen, Universitetsparken 2, Copenhagen 2100, Denmark. [5] European XFEL GmbH, Holzkoppel 4, 22869 Schenefeld, Germany. [6] Present address: Faculty of Pharmacy, Institute of Chemistry, San Pablo CEU University, Urb. Montepríncipe, 28668 Boadilla del Monte, Spain. [7] Present address: Institute of Molecular Systems Biology, ETH Zürich, Zürich, Switzerland. Correspondence and requests for materials should be addressed to T.P. (email: thomas. peters@uni-luebeck.de)

I nfection with human norovirus is the leading cause of acute gastroenteritis worldwide. Attempts to provide antivirals or vaccines have not been successful so far. This is not surprising as noroviruses undergo fast epochal evolution with immune escape variants emerging every 2–5 years[1–4] making it very difficult to design broadly active vaccines or to identify sites on the viral coat protein as targets for entry inhibition. So far, only few studies have focused on the development of such inhibitors[5–8], and the lack of human norovirus cell culture systems has been a major obstacle in proceeding beyond the stage of hit discovery. During recent years two cell culture systems have been developed[9–11] and, in the future, will allow testing of entry inhibitor candidates in cell culture-based infection assays. Binding of human norovirus to histo blood group antigens (HBGAs), a critical step preceding host-cell entry, has been the focus of a substantial number of biophysical studies involving nuclear magnetic resonance (NMR) experiments, native mass spectrometry (MS), surface plasmon resonance, or crystallography as this has been reviewed recently[12–15]. The highly conserved HBGA-binding sites are located in the P2 subdomain of the P-domain of the norovirus capsid protein VP1[16–18]. Expression of the P-domain in *Escherichia coli* provides homodimeric P-domain species, so called P-dimers[18] that are amenable to HBGA-binding studies. Recently, we observed that HBGA binding to Saga *GII.4* P-dimers yielded discontinuous binding isotherms strongly suggesting the presence of allosteric effects[19]. Initially, we had linked this complex binding behavior to sequential binding of HBGAs to the canonical L-fucose-binding sites and to two additional L-fucose sites that had been identified by crystallography[20] for *GII.10* P-dimers. However, docking experiments[21] and saturation transfer difference (STD) NMR titrations using virus-like particles of *GII.4* and *GII.10* human noroviruses[22] are inconsistent with this interpretation. Therefore, other factors must cause the observed discontinuous binding isotherms.

To shine more light on HBGA binding to human noroviruses we have applied chemical shift perturbation (CSP) experiments[23] to Saga *GII.4* P-dimers. Measurement of CSPs of backbone NH resonances upon ligand binding readily identifies amino acids directly involved in ligand interactions. At the same time, CSP experiments can provide long-range effects that may indicate allosteric networks. CSP titration experiments also provide dissociation constants, and can in favorable cases yield access to binding kinetics. However, to take full advantage of CSP experiments two requirements must be met. First, a high-resolution structure of the target protein of interest must be available, and second, an NMR assignment of all or at least most backbone NH resonances must exist. While high-resolution crystal structures of Saga *GII.4* P-dimers were available[17] the assignment of backbone NH resonances for the 73 kDa P-dimer was a substantial challenge[24]. Our previous protein-based CSP study made use of unassigned NH signals undergoing changes upon ligand titration[19] to determine dissociation constants $K_D$ for L-fucose, the minimal ligand for *GII.4* norovirus. For lack of an assignment this study provided no information about the amino acids involved in binding. Here we present an almost complete assignment of NH backbone resonances of *GII.4* Saga P-dimers. Unexpectedly, the NMR assignment exposed a highly specific deamidation of amino acid N373, resulting in an isopeptide linkage and causing a dramatic decrease of HBGA-binding affinity. These NMR results were supported by crystallography and hydrogen/deuterium exchange MS (HDX MS). HDX MS measures the exchange of protein backbone hydrogens to deuterium in solution. Alterations in the hydrogen bonding network introduced by variable conditions, such as PTMs or ligand interactions, result in altered H/D exchange rates, which makes HDX MS a valuable tool for mapping protein–ligand interaction surfaces and changes in protein

dynamics[25,26]. Sequence analysis of *GII.4* strains suggests that the modification is highly abundant, implying that previously published HBGA binding data may require careful revision. Obviously, such a dramatic change in HBGA binding will have implications for the infection process and, due to the antigenicity of the affected region[27–29], very likely also for the immune response.

The present study builds on our previous work[19] where we monitored binding of methyl-α-L-fucopyranoside to uniformly $^2H,^{15}N$-labeled *GII.4* Saga P-dimers. CSP titration experiments based on unassigned $^1H,^{15}N$ Transverse Relaxation Optimized Spectroscopy (TROSY) Heteronuclear Single Quantum Coherence (HSQC) spectra had yielded discontinuous binding isotherms suggesting cooperative binding. Here, we present an almost complete backbone assignment of the P-dimers and uncover spontaneous deamidation of asparagine 373, influencing HBGA binding. In light of this evidence a simple one-site binding model is plausible.

## Results

**Backbone assignment of *GII.4* Saga P-dimers**. At a molecular weight of 73 kDa for the homodimer backbone assignment using triple-resonance three-dimensional (3D) NMR experiments was challenging. It turned out that sample preparation was the most critical factor for acquiring high-quality 3D NMR spectra of [$U$-$^2H,^{13}C,^{15}N$] labeled P-dimer samples. Long delays in respective pulse sequences are incompatible with short transverse relaxation times ($T_2$) of backbone NH signals. Therefore, increasing protein concentrations improve signal-to-noise (S/N) ratios in triple-resonance 3D NMR spectra only to the point where protein aggregation and associated slower molecular tumbling become dominant. In the case of Saga P-dimers balancing between high protein concentration for better S/N and sufficiently long $T_2$ values was rather delicate (Supplementary Table 5 and Supplementary Figs. 7, 8). To better understand P-dimer aggregation and precipitation, we determined molecular correlation times $\tau_c$ employing $^1H,^{15}N$-TRACT experiments[30]. With increasing P-dimer concentrations we observed an increase of $\tau_c$ and concomitantly a decrease of cumulative $H^N$ $T_2$ values, strongly suggesting formation of aggregates at protein concentrations higher than ca. 200 μM (for details cf. Supplementary Note 2, Supplementary Figs. 5–8, and results from Dynamic Light Scattering (DLS) measurements, Supplementary Fig. 9). $^1H,^{15}N$ TROSY HSQC spectra experiments also showed that a substantial fraction (ca. 30%) of NH backbone signals was missing due to very slow exchange of backbone deuterons $D^N$ to $H^N$. Therefore, to achieve a full backbone assignment an unfolding-refolding protocol allowing for complete $D^N/H^N$ exchange was required (for details cf. Supplementary Note 1, Supplementary Figs. 1–4, and Supplementary Tables 1–4). Following a refolding-screening scheme proposed previously[31], we established an optimized unfolding-refolding procedure that provides excellent yields of refolded P-dimers. $^1H,^{15}N$ TROSY HSQC spectra of refolded [$U$-$^2H,^{15}N$]-labeled samples demonstrated complete $D^N/H^N$ exchange (Fig. 1a, b). Notably, Fig. 1b and Supplementary Fig. 4 show that slowly exchanging $H^N$ are not only located in β-sheets or α-helices, where they would be expected. For example, slow $D^N/H^N$ exchange is also observed for the stretch of amino acids G264 to S279, forming an extended loop region, which is stabilized by a network of intra-loop hydrogen bonds as seen from the crystal structure (pdb: 4X06). Backbone $H^N$ involved in respective intra-loop hydrogen bonds are invisible or have significantly reduced intensities in non-refolded samples, reflecting protection and likely reduced flexibility of this region of the protein.

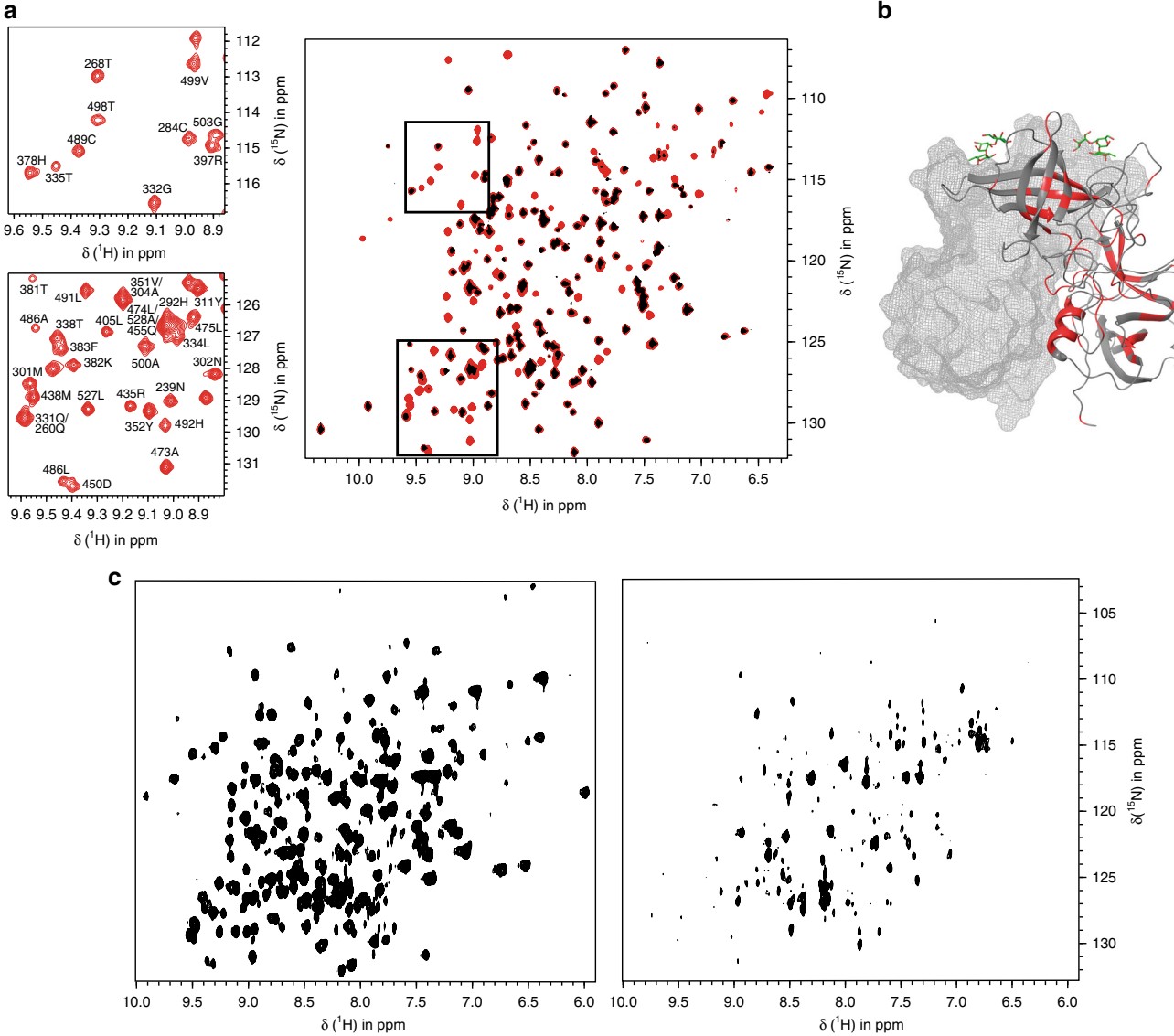

**Fig. 1** Exchange of backbone NH protons and protein concentration affect spectral quality. **a** $^1$H,$^{15}$N TROSY HSQC spectra of 100 µM [$U$-$^2$H,$^{15}$N] labeled *GII.4* Saga P-dimers before (black) and after (red) protein refolding recorded on a Bruker AVIII 500 MHz NMR spectrometer. Samples were prepared in buffer containing 75 mM sodium phosphate, 100 mM NaCl, and 100 µM DSS-$d_6$ in 10% $D_2O$ with the pH* adjusted to 7.30 and with the temperature set at 298 K. **b** Non-exchangeable NH protons (red color) mapped onto the surface of *GII.4* Saga P-dimers (pdb: 4X06). **c** Positive $^1$H-$^{15}$N projection of all planes along the $^{13}$C dimension of 3D TROSY HN(CO)CACB spectra acquired under identical conditions (in sodium phosphate 20 mM, pH* 7.30) for [$U$-$^2$H,$^{13}$C,$^{15}$N] *GII.4* Saga P-dimers at concentrations of 180 µM (left) and of 300 µM (right). Source data are provided as a Source Data file

Misadjusting experimental conditions has a dramatic effect on spectral quality. This is shown in Fig. 1c comparing the $^1$H-$^{15}$N projection of all planes along the $^{13}$C dimension of a HN(CO)CACB experiment for two concentrations of refolded P-dimers. An optimal concentration of 180 µM is compared to a slightly increased concentration of 300 µM.

For the backbone assignment we followed established strategies employing triple-resonance 3D NMR experiments[32]. Using a [$U$-$^2$H,$^{13}$C,$^{15}$N]-labeled sample of refolded P-dimers we were able to assign 86% of all NH backbone resonances. Except for one case all missing assignments are within surface-exposed loops suggesting local conformational interconversion on the intermediate chemical shift timescale. A BMRB[33] query yielded only three outliers of $^{13}$C chemical shifts of C$^\alpha$ and C$^\beta$ atoms from the expected distribution, i.e. the deviation from the mean chemical shift was >3$\sigma$. In each case (C$^\beta$ of Q336, C$^\beta$ of A349, and C$^\beta$ of P400) the

deviation can be explained by the presence of neighboring aromatic side chains. Validation of the chemical shift assignment was accomplished by predicting secondary structure elements exclusively based on the observed chemical shifts using the TALOS-N algorithm[34]. Except for very few positions, likely due to missing assignments or signal overlap, predicted secondary structure elements match very well with those observed in the crystal structure (see Supplementary Fig. 10).

**NMR reveals an irreversible post-translational modification.** During the assignment process we noticed an irreversible transformation of P-dimers into a second species. Over time, new cross peaks appeared and other peaks disappeared in $^1$H,$^{15}$N TROSY HSQC spectra. Complete conversion was observed independent of the presence or absence of HBGAs but was significantly affected by the temperature. It is well established that peptides

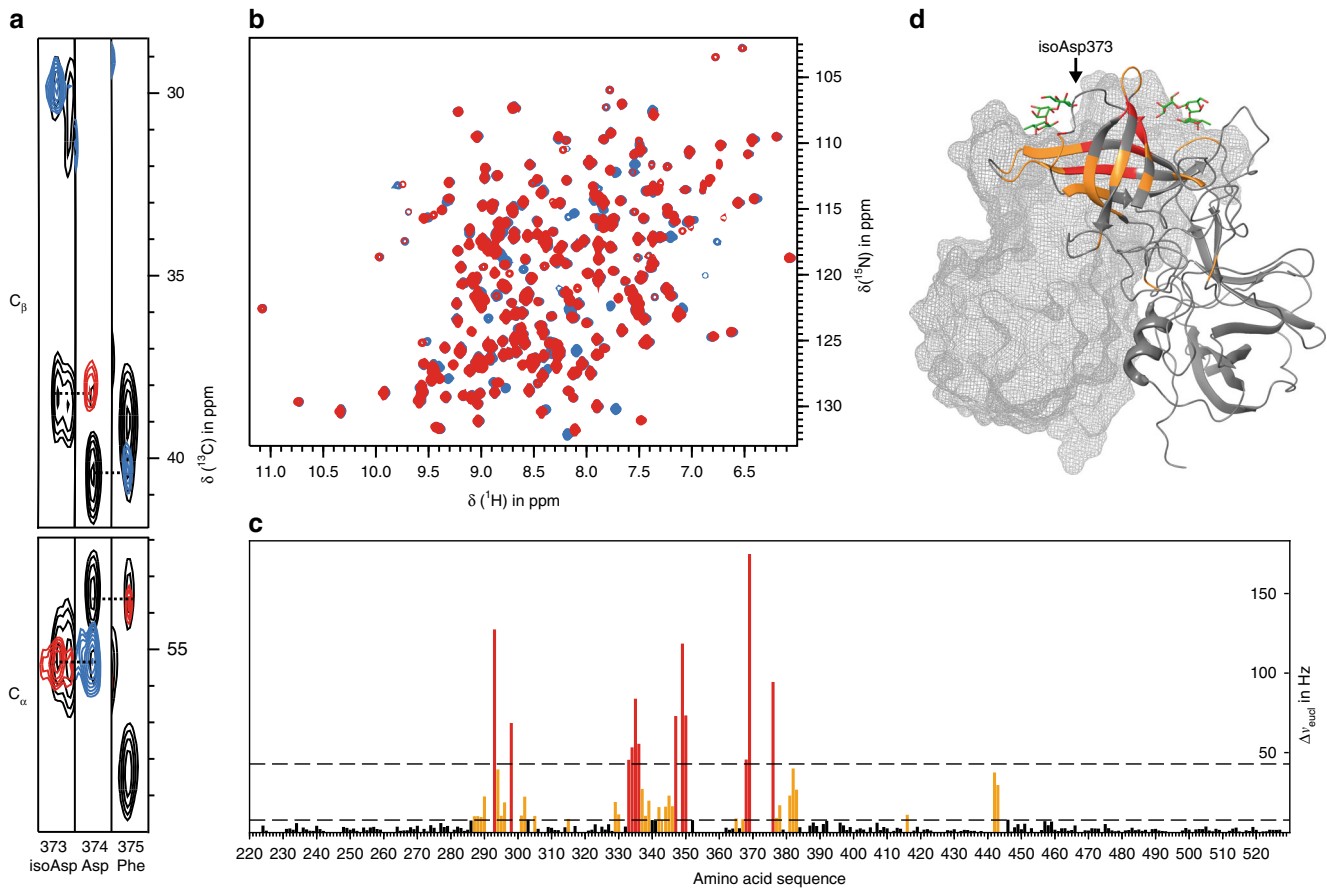

**Fig. 2** Three-dimensional NMR spectra reveal deamidation of N373 and formation of isoD373. **a** Slices from HNCACB (black) and HN(CO)CACB spectra (red and blue) at the $^1H^N$ and $^{15}N$ chemical shifts of isoD373, D374, and F375. The cross peak at the $^{13}C^\alpha$ chemical shift of D374 is negative (blue) showing that the peptide bond is now via the side chain of the deamidated Asn residue in position 373 yielding an iso-Asp (isoD373). **b** $^1H,^{15}N$ TROSY HSQC of native Saga P-dimers (red) and deamidated, post-translationally modified Saga P-dimers (blue). **c** Chemical shift perturbations (CSPs) induced by irreversible deamidation of N373 with dashed lines at $2\sigma$ (red color) and at the experimentally determined significance threshold of 7.9 Hz (orange color, see Supplementary Note 2 and Supplementary Fig. 11 for details). **d** CSPs mapped onto the crystal structure of P-dimers complexed with B trisaccharide (pdb: 4X06). N.B.: for the native P-dimer containing N373, an assignment of the loop comprising amino acids 370–375 is missing. Source data are provided as a Source Data file

**Fig. 3** Deamidation of Asn373 and formation of isoD373. Nucleophilic attack of the side-chain carboxylic acid carbon of Asn373 by the backbone nitrogen of Asp374 leads to formation of a reactive succinimide derivative via a nucleophilic acyl substitution. The unstable succinimide reacts with water molecules to yield either isoAsp373 (isopeptide bond) or Asp373 (normal peptide bond). Red bonds highlight the peptide backbone

and proteins undergo a variety of post-translational modifications, in particular covalent attachments of glycans or phosphate groups to amino-acid side chains. It is less noticed that PTMs can also alter the protein backbone, although very prominent examples exist[35]. A common but often overlooked backbone PTM is the formation of isopeptide bonds via deamidation and subsequent rearrangement of asparagine residues. Following this line of thoughts, we inspected sequential connections in 3D TROSY HNCACB and HN(CO)CACB spectra, which directly report on the formation of isopeptide linkages[36,37] by a phase change of cross peaks sequentially connecting amino acids. Indeed, a tell-tale phase change was observed in converted samples for the cross peaks connecting N373 to D374. The cross peak at the $H^N$ and $N^H$ chemical shifts of D374 and the $^{13}C^\alpha$ chemical shift of the $i$-1 amino acid, N373 in the native form of the P-dimer, has a negative sign in the HN(CO)CACB spectrum (blue color in Fig. 2a) but should be positive for a conventional peptide bond. This phase change of the cross peak is due to formation of an isopeptide bond connecting the backbone NH of D374 to the side chain $C^\gamma$ of N373 reversing the roles of $C^\alpha$ and $C^\beta$ in the HN(CO) CACB experiment. Therefore, the observed phase change is unambiguous evidence for rearrangement of N373 into an iso-aspartate (isoD373), as was described previously for malate synthase[37]. Of note, the backbone NH resonances of P-dimers containing isoD373 instead of N373 have been assigned, too. In fact, some assignments were only made for the isoD373 form (cf. Supplementary Table 7).

Conversion of N373 to isoD373 follows the mechanism shown in Fig. 3. Nucleophilic attack of the backbone NH of D374 leads to deamidation of N373, yielding a succinimide as the reactive intermediate. Generally, subsequent nucleophilic acyl substitution with water furnishes a mixture of aspartate and iso-aspartate, with iso-aspartate being the main product. As $^1H,^{15}N$ TROSY HSQC spectra of Saga P-dimers do not indicate the presence of D373 isomers formation of an isopeptide linkage in this case seems to be significantly preferred (>90%) over formation of a regular peptide bond. The intermediate succinimide was not detected, which is likely due to its short half-life.

**N373 rapidly converts into isoD373**. For a more quantitative assessment of the conversion of N373 to isoD373 we established an improved purification protocol. We have used ion exchange (IEX) chromatography for separation of isoD373 P-dimers from the native protein since conversion of N373 to isoD373 generates one additional negative charge. The IEX chromatograms exhibit three major peaks, one for the native form with N373 present in both monomeric units, from now on called NN P-dimers. Two additional peaks correspond to the completely converted form, carrying an isoD373 in both monomeric units (iDiD P-dimers), and to the asymmetric form, carrying isoD373 only in one of the monomeric units (iDN P-dimers). Respective IEX chromatograms are shown in Supplementary Fig. 20. A comparison of $^1H,^{15}N$ TROSY HSQC spectra of native and deamidated P-dimers (Fig. 2b) shows that the largest CSPs are found in the vicinity of the site of transformation (Fig. 2c, d) but there are also notable long-range CSPs. The regions with large CSPs are in line with deuterium uptake differences seen in the HDX MS data (see results below). It turned out that pure native NN P-dimers are not available from synthesis in bacteria because deamidation and isopeptide bond formation already occur during protein expression. NMR experiments and IEX revealed that freshly expressed samples always contain the isoD373 form in varying amounts, depending on the expression protocol and, in particular, on the temperature used during bacterial protein synthesis.

The IEX protocol was used to monitor conversion of purified native NN P-dimers into iDiD P-dimers as a function of time yielding information on the half-life of NN P-dimers at different temperatures (Supplementary Fig. 20). At room temperature and at temperatures used for sample storage (277 K) deamidation and subsequent formation of the isoD373 is rather slow with a half-life of several days to weeks. However, at body temperature (310 K) conversion is significantly faster with an estimated half-life of 1.6 days (Supplementary Fig. 21).

The IEX separated native and deamidated *GII.4* Saga P-dimers were also subjected to peptic cleavage followed by liquid chromatography (LC) separation and peptide identification using tandem MS (MS/MS). Peptides containing the potential deamidation site show peaks with clearly distinguishable retention times in the chromatogram and could be assigned to either the pure native or deamidated form, which differ in mass by +0.98402 Da (Supplementary Fig. 24). Analysis of peptides covering the entire protein sequence revealed no other deamidation sites apart from N373 (Supplementary Fig. 26).

We selected P-dimers of three other norovirus strains, *GII.4* MI001[38], *GII.10* Vietnam 026[39], and *GII.17* Kawasaki 308[40] to test whether deamidation of N373 is also relevant for strains other than *GII.4* Saga, and to explore how site-specific this transformation is. *GII.4* MI001 contains an asparagine at position 373, whereas the other two strains lack an asparagine residue at the equivalent position. Samples were incubated at elevated temperatures and IEX chromatograms were obtained before and after incubation as explained in detail in Supplementary Note 4. We observed deamidation for MI001 (Supplementary Fig. 22) but not for the Kawasaki 308 or Vietnam 026 strains.

We also subjected these samples to peptic cleavage followed by LC-MS. For *GII.4* MI001 deamidation of N373 was detected along with minor deamidation of another asparagine residue (either N446 or N448) after incubation at 37 °C (cf. IEX chromatograms in Supplementary Fig. 22 and MS data in Supplementary Table 12 and Supplementary Fig. 25). The other two strains showed no indication of deamidation in the entire protein confirming the results from IEX chromatography.

Our results suggest that asparagine residues preceding the aspartate that is critical for L-fucose recognition (D374 in *GII.4* Saga) will also deamidate (see Supplementary Note 4) if the respective strain displays a similar loop conformation of the amino acids connecting two antiparallel β-sheets underneath the HBGA-binding site (in *GII.4* Saga this is the loop consisting of residues D370-N380, cf. Supplementary Figs. 18 and 19 and Supplementary Table 10).

**NMR experiments demonstrate altered glycan binding**. In order to evaluate the influence of the observed PTM on HBGA recognition we performed CSP NMR experiments using methyl α-L-fucopyranoside and blood group B trisaccharide as representative HBGA ligands. In our previous work[19] we had already studied binding of methyl α-L-fucopyranoside to *GII.4* Saga P-dimers employing chemical shift titrations. Since at that time the backbone PTM had not been discovered yet the analysis was based on a mixture of NN, iDN, and iDiD P-dimers and could not provide pure dissociation constants for either species. Therefore, we repeated a chemical shift titration using methyl α-L-fucopyranoside as a minimal HBGA ligand employing NN and iDiD P-dimers. Chemical shift titrations employing freshly prepared NN P-dimers were performed within <4 days to keep conversion at a minimum. iDiD P-dimers were stable, showing no signs of conversion during titration. As many of the backbone NH resonances are highly sensitive to even very small pH changes (unpublished data) we used imidazole as an internal pH

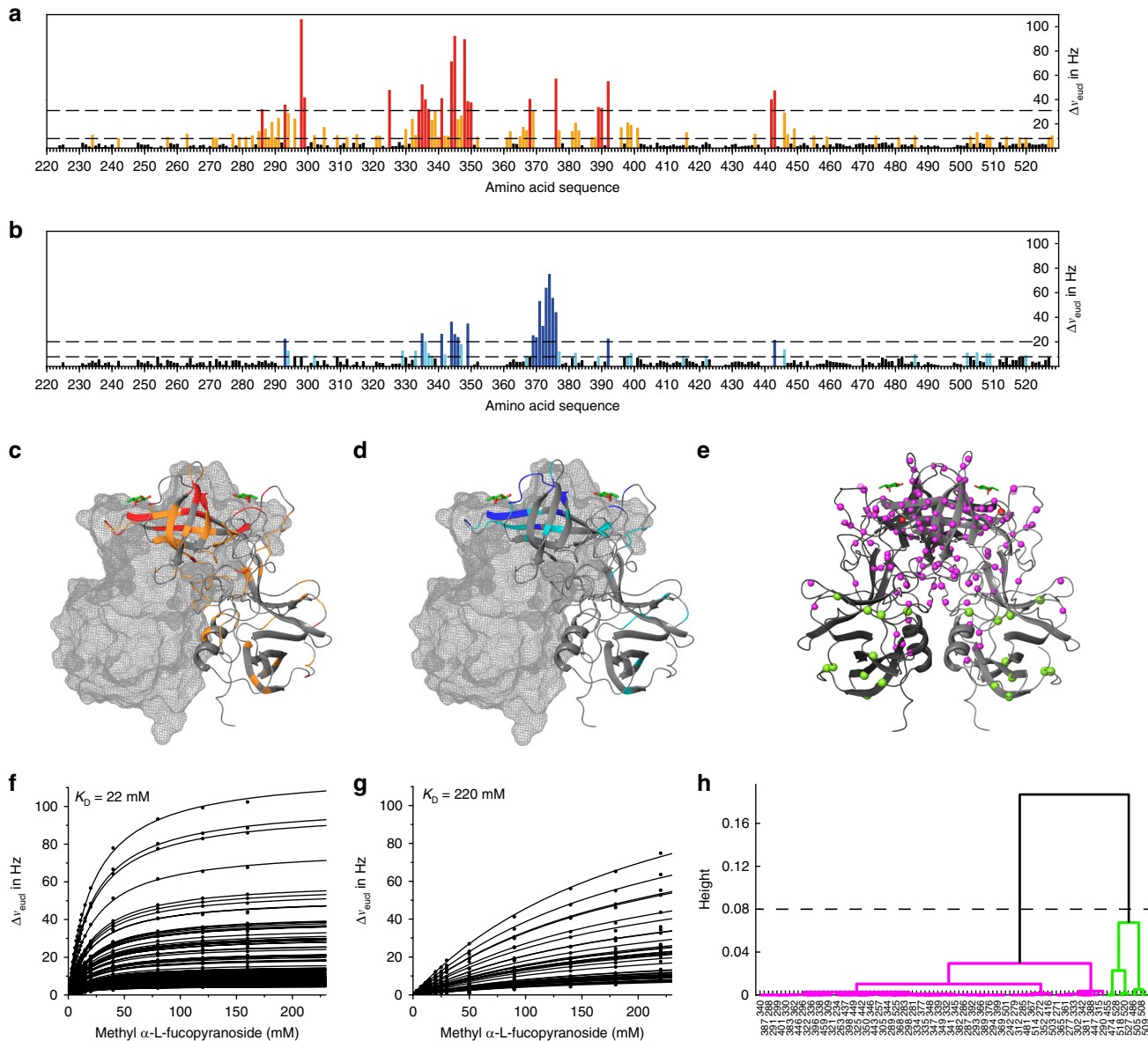

**Fig. 4** Binding of methyl α-ʟ-fucopyranoside to *GII.4* Saga P-dimers. **a**, **b** Chemical shift perturbations (CSPs) of NH backbone signals of *GII.4* Saga NN P-dimers (**a**) and iDiD P-dimers (**b**) in the presence of 160 and 220 mM methyl α-ʟ-fucopyranoside, respectively. Dashed lines are at $2\sigma$ (red, blue) and at the experimentally determined significance threshold of 7.9 Hz (orange, light blue). **c**, **d** CSPs for NN (**c**) and iDiD P-dimers (**d**) mapped on the crystal structure (pdb: 4X06) of *GII.4* Saga P-dimers. Color coding as in **a** and **b**. **e** Clusters of amino acids (N backbone atoms are shown as balls) of NN P-dimers reflecting two distinct types of binding isotherms (cf. Supplementary Note 2 and Supplementary Fig. 13). Color coding is as in **h**. **f**, **g** Global fitting of the law of mass action to chemical shift titration curves of NN P-dimers for amino acids belonging to the magenta cluster (**f**) and for iDiD P-dimers (**g**). For details of the curve fitting see Supplementary Note 2, Supplementary Equation 11, and Supplementary Table 8. The curves reflect one-site binding, and global fitting yields dissociation constants of $K_D = 22$ mM and of $K_D = 220$ mM for NN and iDiD P-dimers, respectively. **h** Complete linkage clustering separating amino acids of NN P-dimers based on distinct shapes of binding isotherms into two clusters (magenta and green). Source data are provided as a Source Data file

standard[41]. Data were denoised using singular value decomposition (SVD) as described before[19], and CSPs were calculated as Euclidean distances. As the resulting curves exhibited visibly different shapes the data were subjected to complete linkage clustering employing the Pearson algorithm (cf. Supplementary Note 2, and Supplementary Figs. 12 and 13 for details), yielding two distinct clusters of binding isotherms for NN P-dimers. Mapping the two clusters onto the surface of the crystal structure (Fig. 4e, h) separates residues in the upper part, which includes the HBGA-binding site, from residues in the bottom part. The binding isotherms of the cluster comprising the binding site residues perfectly match a one-site binding model, and global

fitting of Supplementary Equation 11 yielded a dissociation constant $K_D$ of 22 mM for binding of methyl α-ʟ-fucopyranoside to NN P-dimers (Fig. 4f and Supplementary Table 8). This value is considerably higher than dissociation constants reported in previous work[19,42]. Binding isotherms belonging to the other main cluster were precluded from further analysis at this point. For iDiD P-dimers, a cluster analysis was impossible due to much weaker CSPs and to the inability to reach saturation. Although saturation could not be reached, we estimated the dissociation constant by global fitting of iDiD titration curves as 220 mM, demonstrating a dramatic decrease in affinity upon conversion of N373 into isoD373 (Fig. 4g).

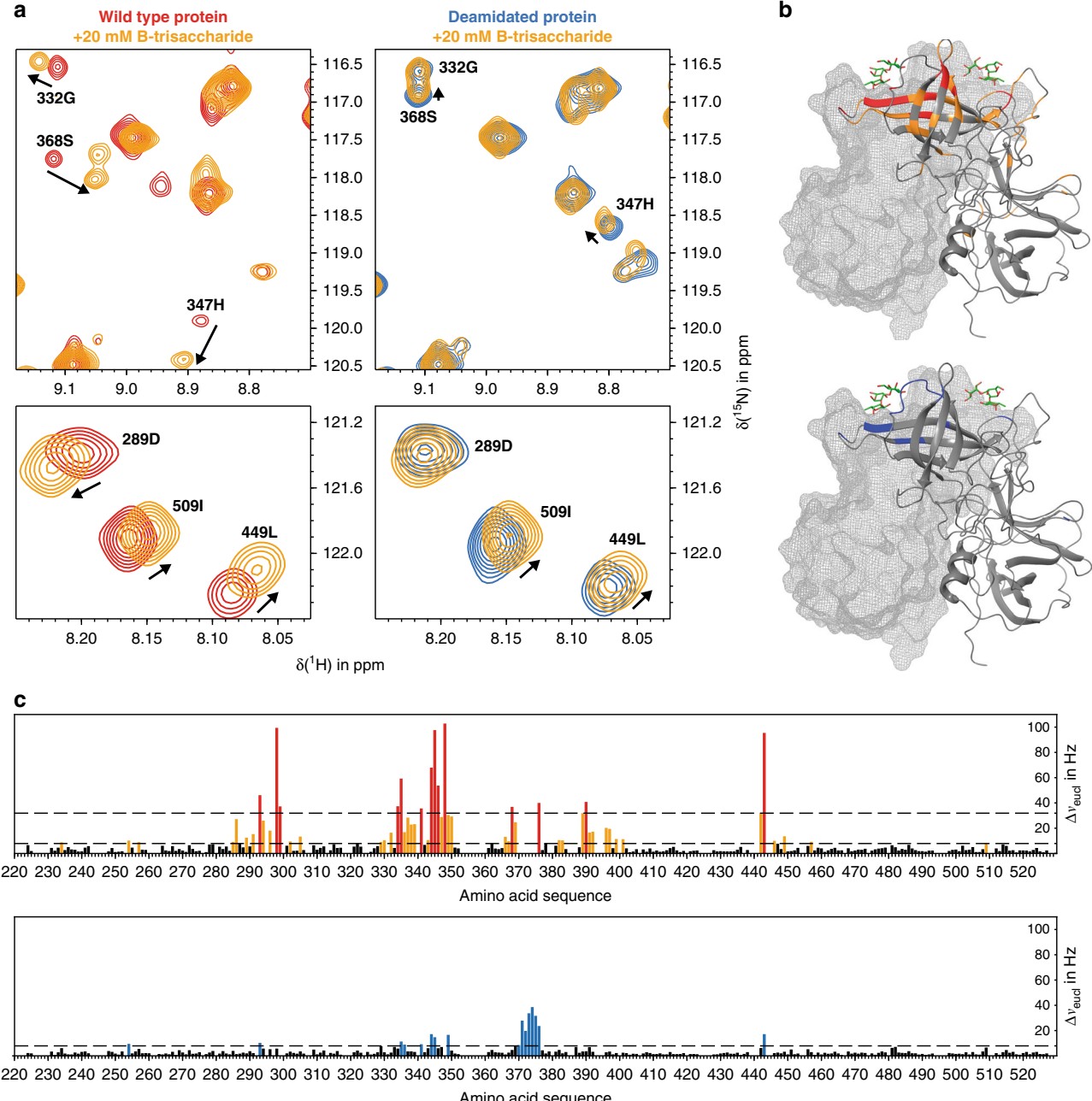

**Fig. 5** Binding of blood group B trisaccharide to *GII.4* Saga P-dimers. **a** ¹H,¹⁵N, TROSY HSQC spectra of native (red, left) and deamidated (blue, right) Saga P-dimers in their free forms and in the presence of 20 mM B trisaccharide (orange). Perturbations induced by ligand binding are indicated by arrows for selected amino acids. **b** Mapping of chemical shift perturbations (CSPs) for both forms onto the crystal structure of P-dimers complexed with B trisaccharide (pdb: 4X06). **c** Complete chemical shift mapping of the respective P-dimer species with experimentally determined 7.9 Hz significance thresholds and additional 2σ threshold to indicate largest CSPs in native P-dimers. Source data are provided as a Source Data file

The deamidation reaction also allowed assignment of the N373 side chain NH as this signal disappeared upon conversion into an iso-aspartate. Not surprisingly, this signal showed by far the largest CSP upon addition of methyl α-L-fucopyranoside, and separate fitting yielded an identical dissociation constant for binding to NN P-dimers (Supplementary Fig. 14).

In order to demonstrate that the loss of HBGA-binding affinity upon transformation of N373 into isoD373 is not confined to methyl α-L-fucopyranoside we present qualitative CSP data on the binding of blood group B trisaccharide as a representative HBGA ligand to NN and iDiD P-dimers. Blood group B trisaccharide was added at a concentration of 20 mM to NN P-dimers and to

iDiD P-dimers. CSPs were observed for iDiD P-dimers demonstrating B trisaccharide binding. A comparison to CSPs observed for NN P-dimers reveals significant differences (Fig. 5). At the same B trisaccharide concentration CSPs are notably larger for NN P-dimers than for iDiD P-dimers indicating a reduced binding affinity for iDiD P-dimers.

Finally, we reanalyzed the published chemical shift titration dataset[19] for binding of methyl α-L-fucopyranoside in the light of our findings (for details cf. Supplementary Note 2, and Supplementary Figs. 15 and 16). Since the published binding isotherms reflect binding to a mixture of NN, iDN, and iDiD Saga P-dimers, we had to exclude all cross peaks resulting from

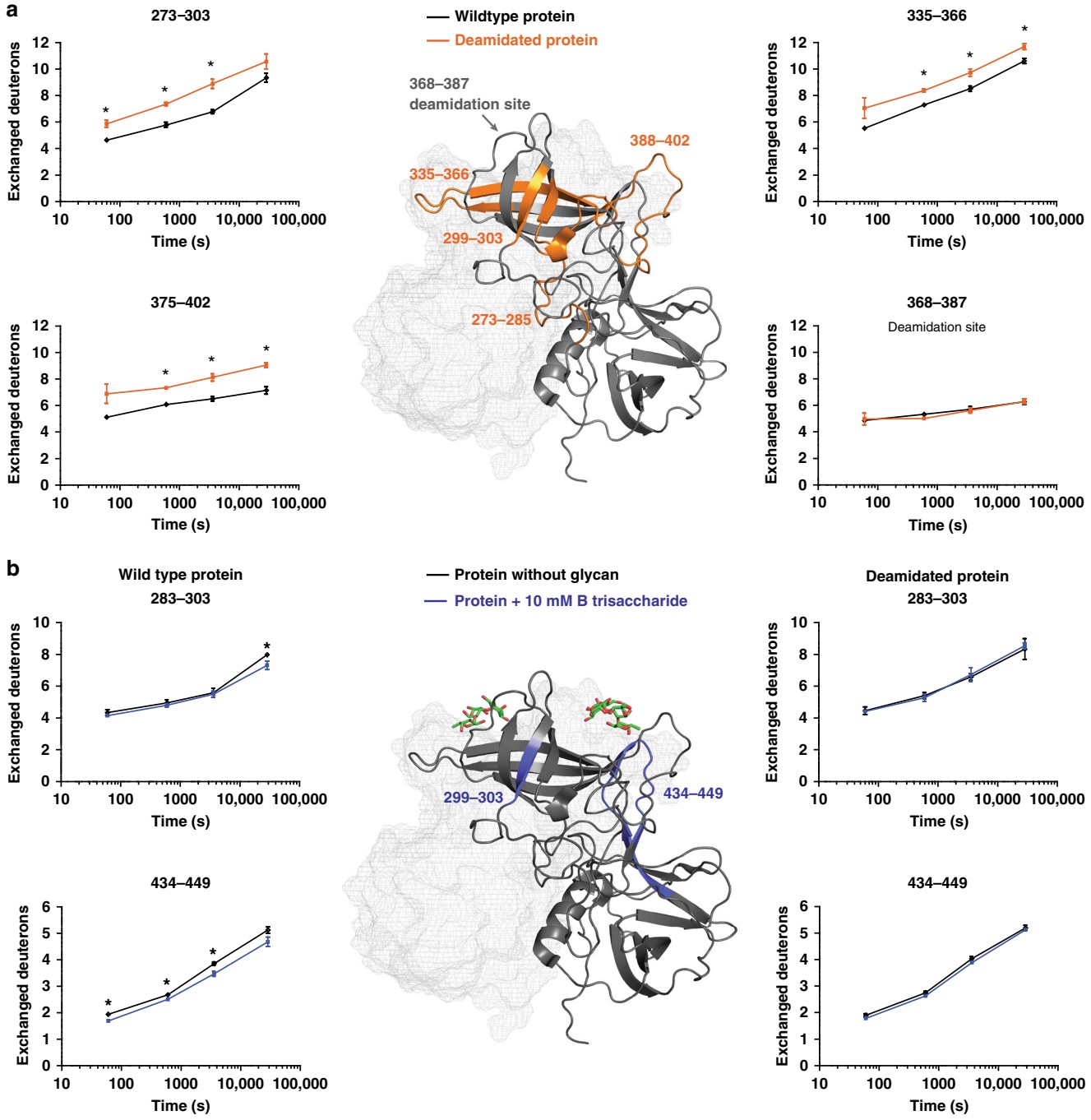

**Fig. 6** Hydrogen/deuterium exchange (HDX) MS experiments reveal changes in protein dynamics upon deamidation. **a** Deuterium uptake differences of the wild-type and deamidated *GII.4* Saga P-dimer mapped to the crystal structure (pdb: 4 × 06). **b** Deuterium uptake differences of the wild-type and deamidated P-dimer in presence and absence of 10 mM B trisaccharide. Significant (*$p < 0.01$, Student's *T*-test) protection is detected in peptides covering the canonical binding site (G443, Y444), whereas this effect is absent in the deamidated P-dimer. The reduction in HDX observed in peptide 283–303 can be narrowed down to residues Y299-L303, as a peptide covering residues I283-N298 shows no differences in exchange at the 8 h time point (see Supplementary Data 1–4 for details). All measurements were performed in triplicate, and error bars indicate the standard deviation. A complete collection of deuterium uptake plots is found in in Supplementary Data 1–4

deamidated species leaving us with binding isotherms reflecting binding to NN P-dimers. Data were denoised using SVD, and Euclidean distances were calculated. Curves with maximal CSPs larger than $2\sigma$ (Supplementary Fig. 16) match a single-site binding model well, and are associated with amino acids located in the vicinity of the fucose-binding site. Global fitting of the law of mass action to the titration curves yields a dissociation constant $K_D$ of 16 mM very similar to the value obtained from the

current titration, with the slightly higher affinity likely due to the use of a different buffer system.

**HDX MS reveals localized alterations in protein flexibility.** To substantiate our findings from NMR experiments, HDX MS was used to map changes in the protein dynamics caused by protein–glycan interactions. Notably, the deamidated P-dimer

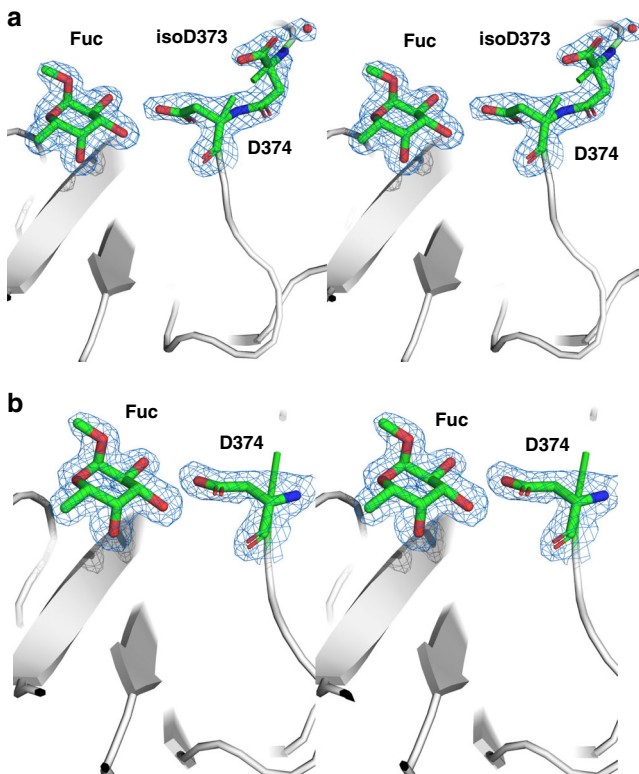

**Fig. 7** Binding of methyl α-L-fucopyranoside to deamidated *GII.4* Saga P-dimers. Wall-eyed stereo representation of the crystal structure showing a non-canonical binding mode of methyl α-L-fucopyranoside in the modified binding pocket. The $2F_O - F_C$ map is shown (σ level = 1.0). **a** For one chain the isopeptide bond (isoD373) and the ligand are clearly visible. **b** For the other chain the loop bordering the binding pocket is unstructured N-terminally of D374, and a canonical binding mode is observed. A corresponding ligand interaction diagram and data collection and refinement statistics are found in Supplementary Note 5 (Supplementary Fig. 23 and Supplementary Table 11)

showed a higher H/D exchange rate in peptides covering large parts of the top P2 subdomain irrespective of the presence of glycans. This indicates higher flexibility of respective regions of the P2 subdomain of the deamidated protein (Fig. 6a and Supplementary Data 3). However, this difference was not detected in the peptide containing the deamidation site. In the presence of 10 mM B trisaccharide the wild-type protein was significantly protected from H/D exchange in peptide 434–449 along the canonical binding site (G443, Y444), indicating occupation of the binding pocket. As seen in the NMR experiments binding is highly reduced in the deamidated protein showing essentially no protection (Fig. 6b). In addition, we detect protection of some residues of a β-sheet in the center of the binding cleft (Y286-L303, Fig. 6b). Moreover, this region contains N298, which displays the largest CSP of all assigned backbone NH resonances (cf. Figs. 4a and 5c), supporting a long-range effect of glycan binding into this region of the protein. Apart from overlapping peptides covering the protein regions presented in Fig. 6 no other peptides of either the wild type (Supplementary Data 1) or the deamidated (Supplementary Data 2) P-dimer show significant deuteration differences, indicating that other subtle long-range effects observed by CSPs, which are extremely sensitive to changes in the chemical environment, are beyond the detection limits of the HDX MS experiments performed here. Binding of 100 mM methyl α-L-fucopyranoside to the wild-type P-dimer results in similar

deuteration differences as observed for 10 mM B trisaccharide suggesting similar interactions. In contrast, no differences can be detected for the negative control, 100 mM D-galactose (Supplementary Data 4).

**Crystallography reveals distinct fucose-binding geometries**. The deamidated form of Saga P-dimers was then subjected to crystal structure analysis in the presence of methyl α-L-fucopyranoside. Well-diffracting apo crystals were obtained belonging to the space group P2₁, different from C2₁ as observed previously for the native form of Saga P-dimers[17]. Upon incubation of deamidated P-dimer apo crystals with 600 mM methyl α-L-fucopyranoside we obtained a co-crystal structure at 1.52 Å resolution, in which the isoD373 isopeptide bond is clearly visible in one of the two protein chains that comprise the asymmetric unit (i.e. in one half of the P-dimer) (Fig. 7 and Supplementary Table 11). We also observed binding of methyl α-L-fucopyranoside to the two canonical HBGA-binding sites in the Saga P-dimer. However, no binding to additional sites (sites three and four) as observed for *GII.10* P-dimers[20] was detectable, even at this drastic ligand concentration. Whereas the overall P-dimer structure is almost identical to previously published Saga P-dimer structures[16] binding of methyl α-L-fucopyranoside to the modified binding pocket shows some deviation from the native P-dimer. In the chain with unambiguous density for the isopeptide bond the side chain of D374 is rotated such that one oxygen of the γ-carboxy group forms a single hydrogen bond to OH-4 of methyl α-L-fucopyranoside (Fig. 7, left). The previously observed binding mode differs somewhat in that a bidentate hydrogen bond involving both oxygen atoms of the γ-carboxy group is formed between the D374 side chain and fucose[16] (corresponding ligand interaction diagrams are shown in Supplementary Fig. 23). This latter orientation is also observed in the second chain in our co-crystal (Fig. 7, right). In this chain, no unambiguous density is observed for residue 373 and the preceding amino acids, and so the density does not allow us to determine whether this chain contains an N373 or an isoD373 residue. Because the crystallization preceded the development of the abovementioned IEX separation protocol, it is possible that the crystallized protein sample contained a certain amount of N373, giving rise to ambiguous density in one of the two chains (where ligand binding is, however, clearly observable).

## Discussion
Our previous studies suggest that binding of human noroviruses to HBGAs involves cooperativity[19,22,43]. Therefore, we set out to better understand underlying molecular principles by studying HBGA P-dimer interactions using protein NMR spectroscopy, HDX MS, and crystallography. We succeeded in obtaining an almost complete NH backbone assignment of the *GII.4* Saga P-dimers using standard 3D triple-resonance NMR experiments, per-deuteration, and an optimized unfolding-refolding protocol for back-exchange of amide protons. This assignment provides an excellent basis for detailed studies into binding of HBGAs as well as other ligands under near-physiological conditions. In the assignment process we discovered that N373, located in a loop directly adjoining the HBGA-binding site, transforms into an iso-aspartate, isoD373, within 1–2 days at body temperature. Experimental proof comes from NMR reflecting the altered connectivity between D374 and N373 via an isopeptide bond by swapping the phases of $^{13}C^\alpha$ and $^{13}C^\beta$ cross peaks of D374 in 3D HN(CO)CACB spectra. This phenomenon was previously described for malate synthase[37] and was applied more recently for the identification of an isopeptide bond connecting an Asn side chain and a Lys side chain of a bacterial surface protein[36]. MS

confirms the mass shift of +0.98402 Da in peptides containing N373/isoD373 supporting the NMR findings.

Since residue 373 is located in a loop critically involved in fucose recognition it is maybe not surprising that the transformation of N373 into an iso-Asp affects HBGA recognition as demonstrated by CSP NMR experiments using methyl α-L-fucopyranoside and blood group B trisaccharide as exemplary ligands (Figs. 4 and 5). In fact, loss of binding affinity due to iso-Asp formation has been described for monoclonal antibodies[44]. Chemical shift titrations show for the case of methyl α-L-fucopyranoside a dramatic loss of affinity of about one order of magnitude upon formation of the isopeptide linkage (Fig. 4). This observation is qualitatively supported by HDX MS experiments showing significantly higher protection from H/D exchange for NN P-dimers as compared to iDiD P-dimers (Fig. 6b). Crystal structure analysis provides an explanation for the observed loss of binding affinity. In our crystal structure of the deamidated P-dimer complexed with methyl α-L-fucopyranoside two binding geometries can be observed at the same time, i.e. one in each crystallographically distinct half of the P-dimer. Fucose binds to the modified binding pocket in much the same place and orientation as observed for the native protein, but the hydrogen bonding to the side chain carboxy group of D374 appears altered as shown in Fig. 7 and Supplementary Fig. 23. The altered D374 side-chain orientation and the loss of one hydrogen bond can be explained by altered conformational dynamics of the loop. In addition, HDX MS reveals alterations of the P2 subdomain flexibility upon formation of the isopeptide linkage. Together, the altered dynamics and consequently less favorable rotamer distribution of the D374 side chain may explain the loss of affinity of the modified binding pocket that is detectable by CSP NMR experiments (Fig. 5). The hypothesis that the isopeptide formation alters the conformational dynamics of the respective loop is based on the observation that NH cross peaks from residues 370–375 of this loop are missing in the native P-dimer TROSY HSQC spectra but are visible for the deamidated form (Supplementary Table 7). Except for the loop carrying the isopeptide bond, the overall structure of the P-dimers is very similar to the structures published before[17]. However, numerous long-range CSPs and deuterium uptake differences measured by HDX MS are observed upon deamidation (Figs. 2d and 6a, respectively). This implies that effects of deamidation are not confined to the vicinity of the deamidation site suggesting differences in overall protein dynamics.

Chemical shift titration analysis shows that binding of methyl α-L-fucopyranoside to NN as well as iDiD P-dimers can be entirely described using a single-site binding model (Fig. 4f, g). Binding isotherms resulting from CSPs of residues belonging to a minor cluster (Fig. 4e–h) are likely secondary effects of glycan binding. Therefore, we reanalyzed our old chemical shift titration dataset[19] to test whether it may match these findings. In fact, global fitting of binding isotherms originating from amino-acid residues in the vicinity of the HBGA-binding site can be analyzed using a single-site binding model, yielding a similar dissociation constant for the binding of methyl α-L-fucopyranoside to NN P-dimers (cf. Supplementary Note 2 and Supplementary Fig. 16). The discontinuities observed in our previous study can now be assigned to amino acids located at remote positions relative to the fucose-binding site, exhibiting weaker CSPs and thus are more sensitive to small variations of experimental conditions. In the light of our data the discontinuities have to be ascribed to variations in sample conditions due to ongoing deamidation (Fig. 3) and the use of a protein concentration of 300 μM, well above the critical limit for aggregation determined in the present study (cf. Fig. 1c). It should be noted that this does not explain the steps observed in STD NMR titration curves[19,22], which

apparently have a different origin. Further work is required to clarify this issue.

Interestingly, recalculation of the dissociation constant $K_D$ for blood group B trisaccharide based on data from a competitive STD NMR titration[19] and using a $K_D$ value of 22 mM instead of 2.4 mM as a reference for methyl α-L-fucopyranoside yields a $K_D$ value of 5.5 mM for blood group B trisaccharide. In the light of our findings we must conclude that our original interpretation and the resulting working hypothesis were incorrect and that HBGA binding is not a cooperative process. However, binding of HBGAs causes many long-range CSPs across distances larger than 20–30 Å. Although the biological implications of these long-range effects are not known yet their presence indicates a rather subtle cross-talk between the HBGA-binding site and other sites in the P-dimer.

What could be the biological significance of irreversible deamidation of N373? Deamidation and isomerization of asparagine residues as well as isomerization of aspartate residues (Fig. 3) are spontaneous processes generating PTMs associated e.g. with protein aging. Therefore, these modifications serve as molecular clocks controlling the life time of proteins[45,46]. On the other hand, a cellular enzymatic repair system exists, reconverting iso-aspartate into aspartate and restoring the conventional protein backbone[47–49] through the action of protein-L-iso-aspartyl O-methyltransferase. Identification and characterization of these PTMs, which are linked to a single unit change in molecular mass, is a challenging analytical problem[50–54]. Therefore, the number of studies describing such modifications and elucidating their biological function is sparse. Nevertheless, there is growing evidence that deamidation of asparagine and isomerization of aspartate are in fact critically involved in a variety of biological processes. For instance, such PTMs may play a regulatory role in apoptosis[55,56] and confer thermal stability in enzymes present in hyperthermophilic bacteria[57]. One study suggested that iso-Asp formation in integrin ligands leads to an isoDGR motif mimicking the canonical RGD integrin-binding motif[58]. In general, there are a number of PTMs modifying the protein backbone and thus modulating protein functions[35]. In the Saga GII.4 P-dimer the site of iso-aspartate formation is in a loop that engages in HBGA binding, and sequence analysis shows that in GII.4 strains 66% of all P-domains carry an asparagine residue in position 373. In fact, almost all GII.4 strains, for which crystal structures of P-dimers have been obtained carry this motif (for details cf. Supplementary Note 3, Supplementary Fig. 17, and Supplementary Table 9). Given a half-life of N373 of the order of only 1–2 days this PTM may play an important role during infection and may also modulate recognition by monoclonal antibodies binding to this region of the P-domain[27–29]. Importantly, vaccines based on subviral complexes presenting P2 subdomains on the surface should be designed such that no deamidation can occur[59]. Correspondingly, screening for anti-virals based on inhibition of HBGA binding and subsequent optimization of binding affinities to generate lead compounds will require employing the non-deamidated form of the protein in respective binding assays. Finally, it will be important to clarify whether deamidation adversely affects the infectivity of nor-oviruses in cell culture systems and thus reduces virus titers.

Which parameters determine the high regioselectivity of the observed PTM? For peptides it is known that the $i + 1$ residue is the main factor controlling the deamidation rate. According to these studies Asn residues followed by a Gly have the shortest half-life[45,49]. It has been recognized early on that for proteins other factors are dominant[60,61]. In particular, the conformation of the sequence containing the deamidation site plays a crucial role. A recent study has addressed this problem by computing free energy changes upon deamidation using a hybrid QM/MM

approach[62]. One important finding is that the deamidation motif requires a defined local conformation of the Asn deamidation site $i$ and the $i + 1$ residue that positions the backbone NH of the $i + 1$ residue at a short distance to the carboxylic acid amide carbon of the side chain of Asn to facilitate nucleophilic attack. Two reactive conformations were identified corresponding to specific combinations of the backbone dihedral angle $\psi$ ($N_i$-$C_i^\alpha$-$C_i'$-$N_{i+1}$) and the side-chain dihedral angle $\chi$ ($C_i'$-$C_i^\alpha$-$C_i^\beta$-$C_i^\gamma$) of $\psi/\chi$ ca. 180°/60° and of $\psi/\chi$ ca. 0°/−60°. Consequently, steric hindrance toward accessing such reactive conformations and solvation of the corresponding transition state are the two single most important factors influencing the rate of deamidation[62]. Analysis of the crystal structures of GII.4 Saga P-dimers (see Supplementary Table 10) shows that N373 in both monomers matches the $\psi/\chi$ ca. 0°/−60° conformation. This reactive conformation is also found in known crystal structures of other GII.4 P-dimers carrying the N373-D374 motif (Supplementary Fig. 18 and Supplementary Table 10). However, another Asn residue in Saga P-dimers, N512, also matches this conformation closely and is followed by a Gly residue, suggesting this site being at high risk for deamidation. In fact, no deamidation is observed for N512 even at prolonged storage at room temperature underscoring that prediction of deamidation sites in proteins based on simple sequence motifs and on local backbone and side-chain conformations is still error prone. This observation may also indicate that the side-chain conformational equilibrium of N512 is altered in solution as compared to the crystal.

Superposition of the loops containing the N373-D374 motif in published crystal structures shows that the conformation of this loop is highly conserved (Supplementary Figs. 18 and 19, and Supplementary Table 10) and, therefore, maybe critical for the formation of a special local environment promoting deamidation. We hypothesize that incorporation of the N373-D374 motif into this loop is necessary and sufficient to induce site-specific fast deamidation. This hypothesis is supported by the observation of deamidation for GII.4 MI001 but not for GII.10 Vietnam 026 and GII.17 Kawasaki 308. The latter two strains display the proper loop conformation but lack an asparagine in position 373. For MI001 the critical loop has the sequence STDTSND[374], which is almost identical to the one found in Saga P-dimers (STDTEND[374]). Although there is no crystal structure available for MI001, this suggests that the conformation of the critical loop is identical to the conformation found for GII.4 Saga.

By revealing a hitherto overlooked backbone PTM in the ligand-binding site of norovirus GII.4 P-dimers this study offers insights into HBGA binding to noroviruses. Our present work demonstrates that binding of methyl α-L-fucopyranoside to P-dimers follows a simple single-site binding model with much higher dissociation constants than reported previously. Formerly observed discontinuities of binding isotherms must be linked to progressive deamidation and protein aggregation. However, binding of methyl α-L-fucopyranoside and of B trisaccharide to P-dimers cause long-range CSPs of amino-acid residues more than 30 Å away from the binding site, indicating the presence of subtle allosteric cross-talk. Future studies are needed to analyze these long-range effects and to explore their biological implications.

An important question arising from our studies is how general isopeptide formation is among different norovirus strains. Simple sequence analysis shows that in GII.4 strains there is a high degree of conservation of N373 of ca. 66%, whereas D374, which is essential for fucose recognition, is 100% conserved. In addition, available crystal structures suggest that the conformation of the loop containing the N373-D374 is highly conserved. Therefore, it is likely that all strains carrying the N373-D374 motif are undergoing the described backbone PTM, which would concern a large fraction of all strains including the predominant ones.

Employing IEX chromatography and MS, it should be possible to systematically test the prevalence of this PTM among different strains. The availability of cell culture systems will then allow infection studies with corresponding strains under controlled conditions and comparative binding to known monoclonal antibodies will inform on the effects of this PTM on immune recognition and immune evasion.

## Methods

**Expression and purification of P-dimers**. GII.4 Saga 2006 (residues 225–530), GII.4 MI001 (residues 225–530), GII.10 Vietnam 026 (residues 224–538), and GII.17 Kawasaki 308 2015 (residues 225–530) P-domains, with GenBank accession numbers AB447457, KC631814, AF504671, and LC037415, respectively, were synthesized and purified using the following protocol. E. coli BL21(DE3) were transformed with a pMal-c2x expression vector encoding the genes for ampicillin resistance, a fusion protein of maltose-binding protein (MBP), two His-tags, a HRV3C cleavage domain, and the P-domain. Due to the cloning strategy, the sequences from GII.4 Saga 2006 and GII.17 Kawasaki 308 2015 P-domains contain an extra GPGS sequence preceding K225, whereas GII.10 Vietnam 026 contains a GPG sequence preceding S224. For synthesis of unlabeled proteins, transformed cells were grown for 3 h at 37 °C in modified terrific broth medium (12 g tryptone, 24 g yeast extract, and 40 ml glycerol per liter culture) supplemented with M9 minimal medium components (0.5 g of NaCl, 3.3 g of KH$_2$PO$_4$, 16.6 g of Na$_2$HPO · 12 H$_2$O, 1 g of NH$_4$Cl, 1 ml of 1 M MgSO$_4$, 1 ml of 0.1 M CaCl$_2$, and 0.2 % glucose per liter culture), 0.4 % casamino acids, and 100 µg ml$^{-1}$ ampicillin. Overexpression was induced with 1 mM isopropyl-β-D-1-thiogalactopyranoside (IPTG) at an OD$_{600}$ value of 1.5. Incubation was continued at 16 °C for 48 h. [$U$-$^2$H,$^{15}$N]- and [$U$-$^2$H,$^{13}$C,$^{15}$N]-labeled GII.4 Saga 2006 P-dimers were synthesized following a modified protocol. Cells were grown in 40 ml of terrific broth medium at 37 °C in an orbital shaker operating at 200 rpm for 16 h. Cells were harvested and resuspended in 40 ml of D$_2$O-based M9+ minimal medium with a starting OD$_{600}$ of 0.1. M9+ medium contained 3 g l$^{-1}$ $^{15}$NH$_4$Cl (Deutero), 3 g l$^{-1}$ deuterated glucose (1,2,3,4,5,6,6-d$_7$, Deutero) for [$U$-$^2$H,$^{15}$N] labeling, or deuterated $^{13}$C$_6$-glucose (Deutero) for [$U$-$^2$H,$^{13}$C,$^{15}$N] labeling as sole nitrogen and carbon sources, respectively. Culture media further contained 13 g Na$_2$HPO$_4$ · 2 H$_2$O, 3.6 g KH$_2$PO$_4$, 1 g NaCl, 465 mg MgSO$_4$, 200 mg MgCl$_2$, and 14.2 mg CaCl$_2$ per l, 100 µg ml$^{-1}$ ampicillin, and 24 % E. coli OD2 DN medium (Silantes), and were supplemented with vitamins (20 mg vitamin B1, 0.1 mg riboflavin, and 1 mg D-biotin, choline chloride, folic acid, nicotinamide, D-pantothenic acid, pyridoxal hydrochloride, and cobalamine, respectively). The culture was incubated at 37 °C until an OD$_{600}$ of 0.5 was reached. Cells were again harvested and resuspended in the final culture volume of fresh M9+ medium. Temperature was lowered to 16 °C at an OD$_{600}$ of 0.8 and expression was induced with 1 mM IPTG. Cells were harvested after an OD$_{600}$ of 5 was reached, resuspended in phosphate-buffered saline buffer, and then lysed using a high-pressure homogenizer (Thermo). The lysate was clarified by centrifugation, and the fusion protein was purified using a Ni-NTA resin (Qiagen). MBP and the His-tag were cleaved from the P-domain using HRV 3C protease (Novagen). Cleaved P-domain protein eluted from Ni-NTA resin and was further purified by size-exclusion chromatography using a Superdex 26/600 75 pg column (GE Healthcare) in 20 mM sodium phosphate buffer (pH 7.3). Protein purity and dimer concentration were monitored by SDS-polyacrylamide electrophoresis and ultraviolet absorption ($\varepsilon_{280}$ 70,820 M$^{-1}$ cm$^{-1}$), respectively. Separation of fully, partially, and non-deamidated P-dimer species was achieved by cation exchange chromatography using a 6 ml Resource S column (GE Healthcare) at 6 °C. Protein samples were prepared in 20 mM sodium acetate buffer (pH 4.9) and eluted using a 78.4 ml linear salt gradient up to 195 mM NaCl with a flow rate of 1 ml min$^{-1}$. The deamidation reaction rate at 310 K of GII.4 Saga P-dimers was determined by incubation of 125 µl P-dimer aliquots with a protein concentration of 2.3 mg ml$^{-1}$ in 75 mM sodium phosphate buffer, 100 mM NaCl (pH 7.3) for up to 48 h and subsequent analytical IEX chromatography. Baseline correction and quantification of peak integrals of the native P-dimer species was performed using Unicorn 7 software (GE Healthcare). GII.4 MI001 and GII.10 Vietnam 026 P-dimers were incubated in 25 mM TRIS, 300 mM NaCl (pH 7.3) at 25 °C for 3 weeks. GII.17 Kawasaki 308 P-dimers were incubated for 48 h at 37 °C in 25 mM Tris and 500 mM NaCl (pH 7.3). All samples were tested for deamidation by IEX chromatography and MS.

**Protein unfolding and refolding**. Conditions for refolding of P-dimers were screened (cf. Supplementary Note 1), and a protocol for D$^N$/H$^N$ exchange was developed. Every step was performed at 6 °C, and materials and solutions used in the process were pre-cooled prior to use. Protein samples were prepared in 20 mM sodium phosphate buffer (pH 7.3) at a protein concentration of 7 mg ml$^{-1}$. One milliliter of the protein solution was transferred into 35 ml of unfolding buffer (4 M guanidine hydrochloride, 0.5 M Tris, 0.3 M NaCl, and 10 mM 2-mercaptoethanol (BME), pH 7.3) and incubated for 4 h. Then, refolding was started by addition of 35 ml of stabilization buffer (3 M guanidine hydrochloride, 0.5 M Tris, 1 M L-proline, 0.4 M D-sucrose, and 10 mM BME, pH 7.3)) to the reaction mixture and subsequent dialysis against 2 l of refolding buffer (0.5 M Tris, 0.5 M L-proline,

0.2 M D-sucrose, and 10 mM BME, pH 7.3) overnight. The protein solution was then dialyzed stepwise against 2 l of diluted refolding buffer (0.25 M Tris, 0.25 M L-proline, 0.1 M D-sucrose, and 5 mM BME, pH 7.3) for 8 h, 2 l of gel-filtration buffer (50 mM Tris, 0.45 M NaCl, and 5 mM BME, pH 7.3) once for 8 h and twice for 24 h. Note, that the volume of the protein mixture doubled during dialysis due to osmotic pressure. Refolded P-dimers were again purified by size-exclusion chromatography with an overall refolding yield of 80%.

**NMR spectroscopy**. All NMR experiments were recorded at 298 K. Spectra were processed using TopSpin 3.5 (Bruker) and analyzed using CCPNMR 2.4.2[63]. NMR spectra for protein backbone assignment were acquired on 900 and 500 MHz spectrometers (Bruker) equipped with a TCI cryogenic probe using TROSY versions of standard 3D pulse programs (for experimental details cf. Supplementary Table 6). Final assignment spectra were acquired with a 180 μM [$U$-$^2$H,$^{13}$C,$^{15}$N] protein sample in 20 mM sodium phosphate buffer, 10 % D$_2$O, and 200 μM DSS-d$_6$ (pH* 7.3). Carbon chemical shift outliers were identified by comparison with the BRMB amino-acid chemical shift database (queried in March 2018) and assignment data were deposited under the accession code 27445. Secondary structure prediction was performed using the TALOS-N webserver[34] with chemical shift corrections for deuterium isotope effects. Predicted structural elements shorter than three amino acids have not been considered for comparison with crystal structure data.

Optimization of sample conditions and ligand titrations were performed on a Bruker AVIII 500 MHz spectrometer equipped with a TCI cryogenic probe. Rotational correlation times, apparent molecular weights, and cumulative $T_2$ relaxation times were estimated employing $^1$H,$^{15}$N-TRACT and spin-echo experiments under a variety of sample conditions (for details cf. Supplementary Figs. 6–8). $^1$H,$^{15}$N TROSY HSQC experiments were acquired with samples in 75 mM sodium phosphate buffer (pH* 7.3), 100 mM NaCl, 100 μM DSS-d$_6$, 8–10 % D$_2$O, and 300–400 μM imidazole and [$U$-$^2$H,$^{15}$N] P-dimer concentrations between 85 and 150 μM unless stated otherwise.

**Ligands used for CSP experiments**. Methyl α-L-fucopyranoside was purchased from Carbosynth. Blood group B trisaccharide α-L-Fuc-(1,2)-[α-D-Gal-(1,3)-]-α-D-Gal-(1,N)-N$_3$ was a gift from Dr. Hanne Peters in our laboratory and had been obtained via enzymatic synthesis from α-L-Fuc-(1,2)-α-D-Gal-(1,N)-N$_3$, which was a gift from Prof. Javier Pérez Castells (CEU San Pablo, Madrid). To reproduce the CSP experiments described here, commercially available blood group B trisaccharide may equally well be used as a ligand.

**HDX MS and peptide identification**. Wild-type and deamidated P-dimer (50pmol) were mixed with glycans at 10-fold of the final concentration and directly diluted 1:9 in 99% deuterated 20 mM Tris buffer (pH 7.4, 150 mM NaCl, 25 °C) to start the exchange reaction. After various time points the exchange reaction was quenched by 1:1 addition of ice-cold quench buffer (300 mM phosphate buffer, pH 2.3, 6 M urea), which decreased the pH to 2.3, and frozen in liquid nitrogen. The wild-type and deamidated datasets with B trisaccharide were acquired on two consecutive days and all time points were performed in triplicate. The wild-type dataset with B trisaccharide, methyl α-L-fucopyranoside, and galactose contains single measurements performed on the same day.

The samples were thawed and injected onto a cooled (0 °C) HPLC System (Agilent Infinity 1260, Agilent Technologies) equipped with a home packed pepsin column (IDEX guard column with an internal volume of 60 μL, Porozyme Immobilized Pepsin beads, Thermo Scientific) in a column oven (25 °C), a peptide trap column (OPTI-TRAP for peptides, Optimize Technologies), and a reversed-phase analytical column (PLRP-S for Biomolecules, Agilent Technologies). Pepsin digestion was performed online at a flow rate of 75 μl min$^{-1}$ (0.23% formic acid in water) and peptides were trapped in the trap column. Peptides were eluted and separated on the analytical column using a 7 min gradient of 8–40% solvent B (solvent A: 0.23% formic acid in water, solvent B: 0.23% formic acid in acetonitrile) at 150 μl min$^{-1}$. MS was performed using an Orbitrap Fusion Tribrid in positive electron spray ionization MS only mode (Orbitrap resolution 120 K, 4 microscans).

Peptide and PTM identification was performed on non-deuterated samples using a longer elution gradient (27 min, 8–40 % solvent B) in data-dependent MS/MS acquisition mode (Orbitrap resolution 120 K, 1 microscan, HCD 30 with dynamic exclusion of Top 20N). A collection of the data identifying deamidation sites is found in Supplementary Note 6 (Supplementary Table 12 and Supplementary Figs. 24–25). Peptide coverage maps for wild-type and deamidated protein are shown in Supplementary Fig. 26.

**Peptide mapping and HDX MS data analysis**. Precursor and fragment ions were searched and matched against a local protein database in MaxQuant[64] with a minimum score of 20 for unmodified and 40 for modified peptides. For peptides carrying the deamidation site spectra were checked manually and chromatographic peak areas were calculated in Xcalibur (Thermo Scientific) giving a wild-type/deamidated peptide ratio.

DeutEx software (obtained from peterslab.org) was used to determine the deuterium uptake. Excel (Microsoft) and GraphPad Prism software (GraphPad Software, Inc.) were used to create uptake plots and perform statistical analysis. For

comparison of triplicate data a Student's $T$-test was used with the α-value set to 0.01. In addition, a peptide was only considered to have a significant HDX difference if overlapping peptides also showed a difference with $p < 0.01$ at the respective time point. For comparison of the wild-type and deamidated protein an additional cutoff of $\Delta D > 0.64$ (99% percentile calculated according to ref. [65]) was used to account for possible day to day variation in the experimental conditions. The peptide coverage map was plotted with MS Tools[66]. To allow access to the HDX data of this study, the HDX Data Summary Table (Supplementary Table 13) is according to the community-based recommendations (Masson, G. R. et al. Recommendations for performing, interpreting and reporting hydrogen deuterium exchange mass spectrometry (HDX MS) experiments; manuscript submitted). HDX data tables as well as MS raw data have been deposited to the ProteomeXchange Consortium[67] via the PRIDE[68] partner repository (dataset identifier PXD011914). Deuterium uptake plots are shown in Supplementary Data 1–4 and HDX data are summarized according to community-based recommendations in Supplementary Note 6 (Supplementary Table 13 and Supplementary Data 5).

**Crystallography**. Crystals of deamidated *GII.4* Saga P-dimers were grown at 293 K by hanging drop vapor diffusion. The protein solution was buffered with 25 mM Tris (pH 7.3) and 25 mM NaCl, and the reservoir solution contained 0.2 M Mg formate (pH 5.9) and 20% PEG (w/v) 3350. Crystals grew in a 1:1 mix of protein and reservoir solution at a protein concentration of 2.5 mg ml$^{-1}$ and improved with seeding from initial crystals obtained without seeding and at a protein concentration of 9.1 mg ml$^{-1}$. For soaking, a solution containing the reservoir solution plus 600 mM of methyl-α-L-fucopyranoside (Carbosynth) was prepared. Upon incubation in this solution, crystals were cryo protected by brief transfer to a drop containing soaking solution, glycerol, and reservoir solution without ligand at a ratio of 5:3:2 before flash-freezing in liquid nitrogen. In all, 2400 diffraction images with 0.1° per frame were collected at the X06SA beamline of the Swiss Light Source in Villigen, Switzerland, and processed with XDS[69].

**Crystallographic model building**. The previously reported Saga P-dimer structure[16] was used for initial refinement against the collected data in *REFMAC*[70]. Difference density maps revealed unambiguous electron density for the methylated fucopyranoside (ePDB ligand description MFU) in both chains of the P-dimer. Water molecules were removed, the ligand was placed manually in the difference map, and the model went through several cycles of real-space refinement in COOT version 0.8.6.[71] and restrained reciprocal refinement using PHENIX[72]. Once electron density had improved to the level where the iso-Asp bond became visible in one chain (chain A in the deposited crystallographic model) it was manually inserted in the peptide chain in COOT (ePDB ligand description IAS). Restraints for the isoD373-E372 and isoD373-D374 bonds were generated with the help of the CCP4 JLigand tool[73]. Final model quality was assessed using MolProbity[74]. A table summarizing data collection and refinement is found in Supplementary Note 5 (Supplementary Table 11).

**Reporting summary**. Further information on experimental design is available in the Nature Research Reporting Summary linked to this article.

## Code availability

Matlab scripts used for data analysis are available from the corresponding author upon request.

## Data availability

Coordinates of the refined crystallographic model and structure factors have been deposited to the protein data bank (pdb) with the accession code 6H9V. NMR assignments have been deposited with the BioMagResBank with the accession code 27445. HDX data tables as well as MS raw data have been deposited to the ProteomeXchange Consortium[67] via the PRIDE[68] partner repository with the dataset identifier PXD011914. HDX MS data are also provided as a Supplementary Data 5 and deuterium uptake plots are provided in Supplementary Data 1–4. Source data for Figs. 2c, 4a, 4b, 4f, 4g, and 5c, and for Supplementary Figures 1a, 1b, 1c, 1d, 1f, 2, 4, 5a, 6, 8, 11, 12a, 12b, 14b, 15, 16, 17, and 21 are provided as a Source Data file. A reporting summary for this Article is available as a Supplementary Information file. All other data supporting the findings of this study are available from the corresponding author on reasonable request.

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

## Acknowledgements

T.P., B.S.B., T.S., and C.U. thank the Deutsche Forschungsgemeinschaft (DFG) for grants Pe494/12–1, BL1294/3–1, STE 1463/7–1, and UE 183/1–1, respectively (all three FOR2327, ViroCarb). We would like to thank Dr. Thorsten Biet for support with the NMR experiments, Dr. Hanne Peters for enzymatic synthesis of blood group B tri-saccharide, and Prof. Javier Pérez Castells for providing us with blood group H dis-accharide. We are grateful for support within the iNEXT program (PID 1483 and PID 2254) giving us access to the NMR high-field facilities at the Bijvoet Center in Utrecht. In particular, we would like to thank Dr. Hans Wienk from the Bijvoet Center for assisting us in any respect. We thank Dr. Mario Schubert (University of Salzburg, Austria) for helpful discussions. J.D. and C.U. would like to thank Prof. H. Schlüter (UKE, University of Hamburg, Germany) for access to high-resolution mass spectrometers and Dr. Daniel Kavan, Dr. Petr Man, and Dr. Alan Kádek for providing the DeutEx software. J.D. acknowledges funding from COST BM1403 and FOR2327 ViroCarb. C.U. acknowledges funding from the Leibniz Association through SAW-2014-HPI-4 grant.

The Heinrich-Pette-Institute, Leibniz Institute for Experimental Virology is supported by the Freie und Hansestadt Hamburg and the Bundesministerium für Gesundheit (BMG). We thank Dr. Grant Hansman (University of Heidelberg, Germany) for providing us with the plasmids of the P-domains of GII.4 Saga, GII.10 Vietnam 026, and GII.17 Kawasaki 308. Prof. Dr. Stefan Taube (University of Lübeck, Germany) is thanked for providing the plasmid of the P-domain of GII.4 MI001. J.M.O. thanks to Fundación Universitaria San Pablo CEU for a mobility grant.

## Author contributions

A.M., R.C., J.D., B.S.B., C.U., and T.P. designed experiments. A.M., R.C. and L.L.G. performed the NMR experiments, and A.M., R.C., L.L.G., and T.P. analyzed and interpreted the NMR data. A.M. and R.C. expressed and purified the P-domains. A.M. established the unfolding-refolding protocol and wrote Matlab scripts for data analysis. R.C. established the IEX protocol and analyzed deamidation rates. J.M.O. expressed and purified Kawasaki 308 P-domains. J.D., E.T., and K.D.R. performed the MS experiments and J.D., E.T., K.D.R., and C.U. analyzed and interpreted the MS data. P.H.O.M. and B.S.B. performed the X-ray experiments, and P.H.O.M., T.S., and B.S.B. analyzed and interpreted the corresponding data. B.S.B., C.U., and T.P. wrote the paper. All authors contributed to data interpretation and commented on the manuscript.

## Additional information

**Competing interests:** The authors declare no competing interests.

