## [Peer Review File · Nature Communications]

Reviewers' comments:

Reviewer #1 (Remarks to the Author):

This manuscript by Mallagaray et al by using proton-NMR, mass spectroscopy, and crystallography report a novel finding that one of the residues (N373) in the P domain of GII.4 norovirus capsid protein is susceptible to spontaneous post-translational deamidation modification. As this residue is in the site where the P domain binds to histo-blood group antigens (HBGA), they further show that this modification significantly influences (diminishes) HBGA binding affinity. The potential significance of these studies stems from the fact that the studies are done with the P domain of GII.4 noroviruses which are the most predominant strains in the global epidemics, and that HBGA is the glycan that these viruses interact with for initial attachment.

The manuscript is fairly well written, and the results are interesting. All the experiments appear to be well done. Although there are novel results and insights from these studies, the significance is rather marginal with limited impact.

In this manuscript, authors revisit (and in fact correct) their own previous contention (ref 18), together with another group which carried out crystallography of P domain of one of the GII.4 strains in complex with methyl α -L-galactopyranoside (ref 19), that discontinuous binding isotherms observed by NMR upon glycan binding indicate a sequential and cooperative binding of HBGAs to multiple fucose binding sites. They now show here that previously observed binding isotherms are due to progressive deamidation and protein aggregation, and that α -L-fucopyranoside to GII.4 P-dimers follows a simple single-site binding model (which was what others had contended and thought).

The results showing that N373 in the bacterially-expressed P domain of this particular strain of GII.4 is rather interesting and perhaps novel too, and this observation does suggest that interpretation of the HBGA binding affinity should take into account the possibility of this effect. However, it is not clear whether this could be generalized. Although ND motif is conserved in about 60% percent of the GII.4 strains as the authors suggest, the possibility such deamidation may not occur because of the local conformations and/or other residue changes in the vicinity cannot be ruled out without further experimentation.

Considering that even with such progressive deamidation the P domain does bind to the glycan in the expected site, as shown by the crystal structure in this paper, the questions are: 1) whether such

deamidation would have any discernible effect on infectivity and susceptibility, and 2) whether such deamidation would even occur in the context of natural infection.

Reviewer #2 (Remarks to the Author):

The manuscript by Mallagaray et al describes the structure and ligand interactions of the norovirus P-domain. This work represents a thorough examination of the P2 -subdomain and its interactions with host cell glycans using several approaches. The authors discover a spontaneous deamidation leading to an isoaspartate residue that directly impacts ligand binding. This discovery addresses previous confounding observations of structural and binding studies with this protein. Overall the study is well conducted, well written, and has an impressive amount of data. There are a few minor misinterpretations of the H/D exchange data, however these are relatively minor and do not impact the overall conclusions or the breadth of the study (see below). The deamidation is also likely to have important implications for vaccine efforts, which I think the authors should include in their discussion.

Major points:

Pg 18, “which may reflect delayed allosteric interactions ... “. I’m not sure what the authors are speculating here. The fact that a difference is seen in the H/D data at the long incubation time has nothing to do with late-onset allosteric effects. The protein’s dynamic state is pre-determined and constant throughout the deuterium labeling process. The difference in deuterium incorporation after 8 hours of labeling simply means that the amides in that peptide that are affected by the glycan binding are the ones that are already very protected and only exchange at long times. This should be revised to simply state that the structure or dynamics of the beta sheet in this region of the protein is also affected by glycan binding.

Overall, I agree that the NMR data provides evidence for some subtle allosteric effects, but the key word here is subtle. If they were significant then I would have expected to also see changes from the H/D exchange data. As to whether the subtle allosteric effects have any biological relevance, who knows. For this reason, I recommend mentioning the long-range effects seen by CSPs, but not focusing the paper as much on this.

Pg 18, last two sentences "...and thus no long-range effects as seen in NMR".

H/D only probes the exchangeability of amide protons that exchange in a relative narrow time window. It is very possible that there are subtle conformational or dynamic changes that are occurring distal to the carbohydrate binding site, but are either too subtle or influence protein motions that H/D does not probe. The chemical shifts by NMR may be a better way to detect these types of subtle changes. I would revise this section to state something like this, rather than speculating that the H/D data shows no difference because different neighboring amides are changing in different directions leading to no apparent change within the full peptide (which I think is pretty unlikely to occur, especially within so many peptides).

Pg 26. "... the question arises, which form should be targeted for the development of antivirals".

If norovirus is reliant on the interaction with HBGAs for entry, and the spontaneous deamidation hinders this, then what is the use in an antiviral targeting the deamidated form? To me it seems obvious that antivirals should target the native (not deamidated) form. If there is something I'm missing here then please include it in this paragraph.

However, the fact that the deamidated protein is presumably less active/infective has other biologically relevant implications. The P2 subunit is being developed as a subunit vaccine, and if it is to elicit an effective immune response then I would expect that it should be designed to avoid any spontaneous deamidation from occurring at N373 (e.g. see pubmed 24718366). Furthermore, could this deamidation explain or contribute to the difficulty in culturing/storing norovirus, since deamidation would likely inhibit its ability to infect over time? I think these points should be included in this paragraph of the discussion. Of course this is all assuming that the deamidation also occurs in the context of the full virus particle, but I think it's still worth making this point.

Figure 1a: I think there is value in analyzing which peaks in the NMR spectra are absent prior to refolding. This would reveal where the very stable hydrogen bonds are. Are these missing peaks consistent with those predicted to be protected by hydrogen bonds in stable secondary structure in the crystal structure? The data is all there and I think it's worth looking at to get further insight into the solution structure/dynamics of the P2 dimer.

Minor points:

Do the authors observe the N373 sidechain amide crosspeaks in the HSQC spectra? This should also disappear in the isoAsp variant so it may be easy to assign, and could provide more information about whether that sidechain is perturbed by carbohydrate binding.

The second line of page 14 should refer to figure 3f (not 3e).

I recommend referring to figure 3g at the end of the first paragraph on page 14.

In figure S20 the deamidation for peptide 368-379 is shown, but this is not a peptide that is shown in the coverage map (fig S21), nor is there H/D exchange data for it. Was this omitted from the H/D analysis due to weak signal? If so then it might be useful to state this in the legend for figure S20.

Reviewer #3 (Remarks to the Author):

The authors present an excellent and wise approach to evaluate and characterize the role of PTM in viruses on receptor binding. A combination of methods, solidly based on NMR, is presented. The topic is very interesting, the results are clearly presented and the conclusions are sound. An important piece of science. Only one reference is missing (on the application of NMR to monitor sugar-protein interactions): Chem Comm 2018, 54(38):4761-4769. doi: 10.1039/c8cc01444b.

Reviewers' comments:

Reviewer #1 (Remarks to the Author):

This manuscript by Mallagaray et al by using proton-NMR, mass spectroscopy, and crystallography report a novel finding that one of the residues (N373) in the P domain of GII.4 norovirus capsid protein is susceptible to spontaneous post-translational deamidation modification. As this residue is in the site where the P domain binds to histo-blood group antigens (HBGA), they further show that this modification significantly influences (diminishes) HBGA binding affinity. The potential significance of these studies stems from the fact that the studies are done with the P domain of GII.4 noroviruses which are the most predominant strains in the global epidemics, and that HBGA is the glycan that these viruses interact with for initial attachment.

The manuscript is fairly well written, and the results are interesting. All the experiments appear to be well done. Although there are novel results and insights from these studies, the significance is rather marginal with limited impact.

In this manuscript, authors revisit (and in fact correct) their own previous contention (ref 18), together with another group which carried out crystallography of P domain of one of the GII.4 strains in complex with methyl α -L-fucopyranoside (ref 19), that discontinuous binding isotherms observed by NMR upon glycan binding indicate a sequential and cooperative binding of HBGAs to multiple fucose binding sites. They now show here that previously observed binding isotherms are due to progressive deamidation and protein aggregation, and that α -L-fucopyranoside to GII.4 P-dimers follows a simple single-site binding model (which was what others had contended and thought).

We would like to stress that knowledge of the possibility of specific deamidation at a site adjacent to the HBGA binding site is very important for the proper analysis of HBGA-binding data of any kind. Depending on the "history" of individual samples very different results may be obtained, provoking misleading interpretations.

The results showing that N373 in the bacterially-expressed P domain of this particular strain of GII.4 is rather interesting and perhaps novel too, and this observation does suggest that interpretation of the HBGA binding affinity should take into account the possibility of this effect. However, it is not clear whether this could be generalized. Although ND motif is conserved in about 60% percent of the GII.4 strains as the authors suggest, the possibility such deamidation may not occur because of the local conformations and/or other residue changes in the vicinity cannot be ruled out without further experimentation.

Whether or not deamidation in the HBGA binding pocket is a more general phenomenon certainly is an important question. Therefore, we are currently studying the possibility of deamidation for P-dimers as well as for VLPs of different norovirus strains, using predominantly IEX and MS. We have added first data on P-dimers of another GII.4 strain (MI001), a GII.10 strain (Vietnam O26), and a GII.17 strain (Kawasaki 308) to the revised manuscript (see p. 13-14 and p. 28-29 of the revised manuscript; see also chapter 5.2 with Fig. S22, Table S12 and Fig. S25 of the revised supporting information). MI001 P-dimers show deamidation of N373 whereas for the other two strains no deamidation is observed. Based on our findings we suggest that the presence of a specific loop conformation containing the ND motif is necessary and sufficient to cause deamidation of the asparagine residue preceding the "critical" aspartate residue that is essential for HBGA recognition. Interestingly, this loop conformation appears to be conserved in crystal structures in many norovirus strains (see Fig. S18, Fig. S19 and Table S10), suggesting this specific deamidation is a common phenomenon and not confined to specific strains.

Considering that even with such progressive deamidation the P domain does bind to the glycan in the expected site, as shown by the crystal structure in this paper, the questions are: 1) whether such deamidation would have any discernible effect on infectivity and susceptibility,

Recent advances in cell culture models will allow to study the effect of deamidation on infectivity in the future, but we feel this is outside the scope of our present work. It should be mentioned that there are also other potential biological effects that need to be studied such as the impact of this spontaneous PTM on the immune response, vaccine development, or the design of entry inhibitors (see below, comments of the second reviewer).

and 2) whether such deamidation would even occur in the context of natural infection.

As P-dimers protrude from the virus capsids there is no reason why major conformational changes should be observed in P-dimers when assembled into a complete virion. Therefore, it is very likely that deamidation also occurs in complete virions. To experimentally validate this hypothesis, we are currently performing MS analysis of norovirus VLPs. It is essential to realize that the observed deamidation is a spontaneous chemical process that is not confined to a test tube. Therefore, it will occur in any biological environment including cells that are infected by viruses. It will be important to study how external conditions such as pH or salt concentration affect this highly specific deamidation reaction. This is ongoing work in our laboratories.

Reviewer #2 (Remarks to the Author):

The manuscript by Mallagaray et al describes the structure and ligand interactions of the norovirus P-domain. This work represents a thorough examination of the P2 -subdomain and its interactions with host cell glycans using several approaches. The authors discover a spontaneous deamidation leading to an isoaspartate residue that directly impacts ligand binding. This discovery addresses previous confounding observations of structural and binding studies with this protein. Overall the study is well conducted, well written, and has an impressive amount of data. There are a few minor misinterpretations of the H/D exchange data, however these are relatively minor and do not impact the overall conclusions or the breadth of the study (see below). The deamidation is also likely to have important implications for vaccine efforts, which I think the authors should include in their discussion.

Possible implications for vaccine development are an interesting aspect. We have added a paragraph addressing this issue (see p. 27 of the revised manuscript). See also below.

Major points:

Pg 18, "which may reflect delayed allosteric interactions ... ". I'm not sure what the authors are speculating here. The fact that a difference is seen in the H/D data at the long incubation time has nothing to do with late-onset allosteric effects. The protein's dynamic state is pre-determined and constant throughout the deuterium labeling process. The difference in deuterium incorporation after 8 hours of labeling simply means that the amides in that peptide that are affected by the glycan binding are the ones that are already very protected and only exchange at long times. This should be revised to simply state that the structure or dynamics of the beta sheet in this region of the protein is also affected by glycan binding.

We agree with the referee and rephrased this section. In fact, there is an interesting correlation with the largest CSP observed for the backbone NH of N298, which is located precisely in this region of the protein. This supports the observation of long-range effects of glycan binding. The corresponding changes are found on p. 19 of the revised manuscript.

Overall, I agree that the NMR data provides evidence for some subtle allosteric effects, but they key word here is subtle. If they were significant then I would have expected to also see changes from the H/D exchange data. As to whether the subtle allosteric effects have any biological relevance, who knows. For this reason, I recommend mentioning the long-range effects seen by CSPs, but not focusing the paper as much on this.

We are aware that the long-range effects picked up by CSPs are due to subtle changes, and we agree that the biological relevance of this is not yet known. We followed the reviewer's advice and shifted the focus away from possible allosteric effects associated with these long-range effects (see Discussion, p. 26 and Conclusions, p. 29). Nevertheless, we believe that long-range CSPs are indicators for some kind of "information flow" inside the protein, an idea that has surfaced over the past years and has ignited research into the use of CSPs not only as indicators for binding but also as predictors for allosteric interactions. As not enough is known at present, we agree with the reviewer to just report our observations and to avoid speculations. We have modified respective interpretations throughout the manuscript accordingly.

Pg 18, last two sentences "...and thus no long-range effects as seen in NMR". H/D only probes the exchangeability of amide protons that exchange in a relative narrow time window. It is very possible that there are subtle conformational or dynamic changes that are occurring distal to the carbohydrate binding site, but are either too subtle or influence protein motions that H/D does not probe. The chemical shifts by NMR may be a better way to detect these types of subtle changes. I would revise this section to state something like this, rather than speculating that the H/D data shows no difference because different neighboring amides are changing in different directions leading to no apparent change within the full peptide (which I think is pretty unlikely to occur, especially within so many peptides).

We are thankful for this comment as it reminds us not to push away with seemingly contradicting results too fast. It is true that CSPs and HDX detect completely different properties, and, as alluded to above, CSPs are much more sensitive to very small changes in the environment of individual nuclei. Therefore, these subtleties of dynamical and possibly conformational changes of the protein may not be detectable by HDX. We have carefully rearranged and rephrased this paragraph keeping in mind not to exaggerate possible implications of such subtle long-range effects (p. 19 of the revised manuscript: "In addition, we detect protection of some residues of a β -sheet in the center of the binding cleft (Y286-L303, Fig. 5b). Moreover, this region contains N298, which displays the largest CSP of all assigned backbone NH resonances (cf. Figs. 3a and 4c), supporting a long-range effect of glycan binding into this region of the protein. Apart from overlapping peptides covering the protein regions presented in Fig. 5 no other peptides of either the wildtype (Fig. S27) or the deamidated (Fig. S28) P-dimer show

significant deuteration differences, indicating that other subtle long-range effects observed by CSPs, which are extremely sensitive to changes in the chemical environment, are beyond the detection limits of the HDX-MS experiments performed here. Binding of 100 mM methyl α -L-fucopyranoside to the wildtype P-dimer results in similar deuteration differences as observed for 10 mM B trisaccharide suggesting similar interactions. In contrast, no differences can be detected for the negative control, 100 mM D-galactose (Fig. S30).").

Pg 26. "... the question arises, which form should be targeted for the development of antivirals".

If norovirus is reliant on the interaction with HBGAs for entry, and the spontaneous deamidation hinders this, then what is the use in an antiviral targeting the deamidated form? To me it seems obvious that antivirals should target the native (not deamidated) form. If there is something I'm missing here then please include it in this paragraph.

Yes. One would clearly aim at the non-deamidated form of the protein to fight acute infection. We wanted to say that the development of antivirals based on entry inhibition at the stage of screening compound libraries for active species and later on, during optimization of binding affinity, it is important to make sure the protein used in respective assays is not deamidated. We have rephrased this sentence accordingly (p. 27 of the revised manuscript).

However, the fact that the deamidated protein is presumably less active/infective has other biologically relevant implications. The P2 subunit is being developed as a subunit vaccine, and if it is to elicit an effective immune response then I would expect that it should be designed to avoid any spontaneous deamidation from occurring at N373 (e.g. see pubmed 24718366).

This is an important point that has escaped our attention. It actually relates pretty much to the development of entry inhibitors targeting the HBGA binding site. We have added a sentence highlighting this aspect (p. 27 of the revised manuscript).

Furthermore, could this deamidation explain or contribute to the difficulty in culturing/storing norovirus, since deamidation would likely inhibit its ability to infect over time? I think these points should be included in this paragraph of the discussion. Of course this is all assuming that the deamidation also occurs in the context of the full virus particle, but I think it's still worth making this point.

As none of the authors has experience in growing norovirus in any of the recently established cell culture systems we cannot answer this question with confidence, but we think this is an important issue. In collaboration with the lab of Stefan Taube we are currently developing ideas how to test the effects of deamidation in mouse norovirus systems. We have briefly addressed this point in the text (p. 27).

Figure 1a: I think there is value in analyzing which peaks in the NMR spectra are absent prior to refolding. This would reveal where the very stable hydrogen bonds are. Are these missing peaks consistent with those predicted to be protected by hydrogen bonds in stable secondary structure in the crystal structure? The data is all there and I think it's worth looking at to get further insight into the solution structure/dynamics of the P2 dimer.

We have already done this analysis but we did not comment on it in more detail. Interestingly, Fig. S4 shows that there are a number of slowly exchanging backbone NH (absent prior to refolding) that are not part of alpha-helices or beta sheets where one would usually suspect slowly exchanging NH. We have inspected the structure in these regions, and most of these NH are in fact tied up in intra-loop hydrogen bonds. However, there are exceptions such as H378, L278, or G223, where looking at the structure one would expect fast exchange. This suggests that in solution protein dynamics may affect the exchange rate. We have added a short paragraph on p. 7 addressing these points, but it is clear that the explanation of such observations will require much more in-depth studies of the protein, likely also involving extended MD simulations.

Minor points:

Do the authors observe the N373 sidechain amide crosspeaks in the HSQC spectra? This should also disappear in the isoAsp variant so it may be easy to assign, and could provide more information about whether that sidechain is perturbed by carbohydrate binding.

Yes, the reviewer is right. We have observed the disappearance of an arginine side chain in the TROSY HSQC spectra, and, therefore, we have assigned this NH signal to the side chain of N373. In fact, the corresponding cross peak shows by far the largest CSP upon titration with HBGAs. As this assignment was "indirect" we decided to exclude it at this time from presentation and discussion, and rather present it later with systematic titrations of more complex HBGAs (currently performed in our lab). However, reconsidering the story presented in this manuscript we agree with the reviewer that this assignment really belongs here. Therefore, we have added a small paragraph on p. 15 extending the assignment part. We have also included a new supplementary figure (Fig. S14) showing the large CSPs observed upon titrations with L-fucose. Non-linear least squares fitting of Eq. S11 to the titration curve for the sidechain NH of N373 yields a dissociation constant identical with the one obtained from global fitting of all NH backbone titration curves (Fig. 3f and Table S8).

The second line of page 14 should refer to figure 3f (not 3e).

Corrected.

I recommend referring to figure 3g at the end of the first paragraph on page 14.

Done.

In figure S20 the deamidation for peptide 368-379 is shown, but this is not a peptide that is shown in the coverage map (fig S21), nor is there H/D exchange data for it. Was this omitted from the H/D analysis due to weak signal? If so then it might be useful to state this in the legend for figure S20.

Yes, the peptide was omitted due to weak signal. We have rephrased the figure legend accordingly.

Reviewer #3 (Remarks to the Author):

The authors present an excellent and wise approach to evaluate and characterize the role of PTM in viruses on receptor binding. A combination of methods, solidly based on NMR, is presented. The topic is very interesting, the results are clearly presented and the conclusions are sound. An important piece of science. Only one reference is missing (on the application of NMR to monitor sugar-protein interactions): Chem Comm 2018, 54(38):4761-4769. doi: 10.1039/c8cc01444b.

The suggested reference has been included.

REVIEWERS' COMMENTS:

Reviewer #1 (Remarks to the Author):

Authors have made appropriate revisions to my comments and I commend them for that. The manuscript is much better now. What authors have found in this study is tantalizing and novel. (Albeit, in my opinion, still needs further confirmation.

Reviewer #2 (Remarks to the Author):

The authors have addressed the reviewer criticisms nicely and the revisions have strengthened the paper. I recommend publication.